# Make Lectures Match How We Learn: The Nonlinear Teaching Approach to Economics

Peng Zhou

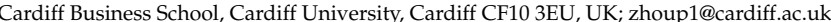

Cardiff Business School, Cardiff University, Cardiff CF10 3EU, UK; zhoup1@cardiff.ac.uk

**Abstract:** This paper proposes a nonlinear teaching approach, based on learning theories in cognitive psychology, with a special focus on large-cohort economics modules. The fundamental rationale is to match the features of teaching with the nature of learning. This approach was implemented in an undergraduate economics module, which received qualitative feedback and quantitative evaluation. Formal econometric models with both binary and continuous treatment effects were developed and estimated to quantify the effects of the proposed approach. Evidence shows that the nonlinear teaching approach significantly improves the effectiveness and efficiency of the learning-teaching process but does not promote student attendance.

**Keywords:** nonlinear teaching approach; higher education; experimental action research; treatment effects

## 1. Introduction

The role of higher education in economic growth is well-recognised in the literature of human capital [1,2]. However, as the direct technology of human capital production, the teaching approach in higher education institutions seems to have evolved slowly, especially for quantitative subjects like economics [3,4]. The COVID-19 pandemic posed new challenges when recorded or online lectures were delivered to large cohorts with limited real-time engagement and feedback [5]. The recent development of generative AI (e.g., ChatGPT) has further transformed higher education and the role of lecturers [6]. To adapt to these disruptive challenges, blended teaching has become popular in universities [7]. Nevertheless, it is usually recognised that existing pedagogies do not always fit students' learning process [8]. Specifically, an important feature of learning process is nonlinearity—learners do not acquire new knowledge in a one-way fashion. To acquire a new piece of knowledge, a typical learner needs to go back and forth many times to eventually understand it and internalise it as part of their knowledge system. New knowledge can be confusing and is easily forgettable, especially when learners are still unclear about the general framework.

To illustrate how people learn, consider a visitor, Sue, who arrives at a new city, say, Cardiff. To know the city, she needs to walk around many times to accumulate information of the roads, districts, the places of interest, and so on. If Sue gets stuck in an unfamiliar corner and gets lost, the most natural choice for her is to try to get back to a place she is familiar with. After some repetition and iterations, a map is formed in Sue's mind. Once a general map is formed, Sue will be able to apply her knowledge of this city to guide her route. Similarly, learning a new theory has the same nonlinear feature. The first level of learning is to become familiar with the knowledge ("city"), corresponding to the process of drawing the map. The second level of learning is to apply the knowledge to practice ("route"). However, the two levels are usually not sequential but iterative. In both stages, learners may make mistakes and forget previous information—becoming "stuck in an unfamiliar corner". The reflection process will help learners to draw a better map and form better applications. An experienced learner may need less time and make less detours

before she acquires the knowledge. Going back to our visitor's example, she may have already travelled to many similar cities before and know what a typical British city looks like. Then, she may get to know the city quite fast within one or two days thanks to her previous experience—"The learners learn how to learn as they learn." [9]

However, conventional teaching practices do not match the nonlinear nature of the learning process well. In some extreme cases, lecturers totally ignore how people learn. A typical lecture tends to be delivered like this: define the key concepts, outline the assumptions, derive the conclusions, and then apply to some popular contexts. As a well-known example in teaching undergraduate microeconomics, lecturers usually define the utility function, outline the maximisation problem, take derivatives, and then apply this information to some consumer problems. A national survey on teaching undergraduate economics was conducted in both 1996 and 2001 by Becker and Watts, who show that the main feature of economics teaching is still "Chalk and Talk" in the US. In contrast to the passive learning environment that characterises the teaching of economics, class discussion and other forms of active learning are now the dominant forms of instruction in other fields of higher education, rather than extensive lecturing [10].

There are two main reasons as to why economics lecturers are reluctant to use alternative teaching methods. On the one hand, economics by nature is a social science, heavily reliant on mathematics and logical derivations. This feature restricts the teaching methods in a passive fashion, similar to natural science disciplines. Active forms, such as discussion, tend to be avoided to achieve efficiency in teaching. However, efficiency does not imply effectiveness. The lecturer may well cover all the technical details of deriving some economic theory from assumptions to conclusions, but the students never know why they are doing these. An even worse outcome is, once you are confused during a lecture, you will never get back on track, because the lecturer is carrying on in a one-way direction. On the other hand, there are insufficient incentives in place to move instructors away from the status quo. For example, lecturers may try not to outperform colleagues and avoid being "too popular" with students to save time for their own research. In economic terms, there are both benefits and costs in teaching innovation, and the equilibrium depends on the optimal balance between the two. The second point can only be resolved by institutional changes in lecturer performance evaluation and is beyond the scope of this paper. Instead, the focus is on the "technological" aspect of improving the effectiveness and efficiency in teaching economics.

To see the drawbacks of linear teaching, let us go back to the visitor example. Assume Sue wants to visit Cardiff Castle from her hotel and she turns to a local resident, Joe, for help. If Joe uses linear teaching, then he may tell Sue to turn left in 100 yards onto North Road, turn right in 200 yards on to Colum Road, then enter the roundabout and take the third exit, etc. Unless Sue is a genius, she will soon get lost and frustrated, because the new road names are easily forgettable. It is not easy to follow all these instructions for those who do not even know where they are standing.

This ineffectiveness of teaching could be due to a lack of the "threshold concept", which is transformative and irreversible [11]. The transformative feature implies that before the learner grasps the threshold concept, she needs to accumulate information piece by piece and an understanding of the knowledge may be partial and erroneous. New information is always easy to forget and confuse. Once the learner has acquired the threshold concept, the general picture becomes clearer, and the learning curve will be steeper. The transformative feature entails a nonlinear teaching process to emphasise and reinforce the threshold concept for the learner. However, the irreversible feature of threshold concept brings difficulty in doing this. For those who have already internalised the threshold concept, they cannot recall how it felt when they did not. For the visitor example, Joe may think it is extremely easy to follow the instructions because he has travelled in that area for years. However, for Sue, it may be extremely difficult to follow because that area sounds like a black box to her. This makes communication difficult because lecturers do not even know where the students are confused.

This paper proposes an approach suitable for undergraduate, large-cohort economics teaching, though this approach can be easily applied to any social science subject at other levels. It is not conventional passive linear teaching (ineffective for economics), and it is not fancy active unstructured teaching (inefficient for economics) either. Rather, it cuts in the middle, which I refer to as the "nonlinear" teaching, because it appreciates the nonlinear nature of the learning process. The notion of nonlinear teaching offers an open system in which interconnected knowledge and sense-making are constantly re-built by dynamic interplays between students' existing knowledge and the new knowledge [12].

In fact, there are some practices in the literature sharing the same rationale of nonlinear teaching proposed here. For example, an early paper by Bartlett and King proposes teaching economics as a laboratory science, with the help of computers and software [13]. This approach emphasises learning-by-doing, i.e., the students are effectively "doing" economics like those in real economics profession. Hadsell proposes to promote students' engagement or active learning by confronting their own values and judgements against the economic theory [14]. This is also one way of sense making, because students only learn something when knowledge is accommodated into their existing system of belief. Other similar examples include Salemi [15] and Chow et al. [16], who promote active learning through group discussion or game play. All these attempts follow either a Behaviourist view or a Constructionist view, which form the starting point of this paper. However, the nonlinear teaching approach will go much further in associating the new theory and skills with the students' existing knowledge, so that the new knowledge can fit into their current knowledge framework. On the other hand, due to the disciplinary restrictions of economics, the proposed approach is not a radical revolution towards, say, totally unstructured open discussions in arts or philosophy. It takes into account the trade-off between effectiveness and efficiency in teaching and learning economics.

The contribution of this paper is to provide a formal theoretical foundation and a systematic teaching approach toolbox. Empirically, both qualitative and quantitative evaluations are conducted through an experimental action research, which has seldom been performed before in the literature. Following this introduction, Section 2 critically reviews the current literature and identifies gaps in large-cohort teaching in higher education. Section 3 discusses the conceptual framework and the theoretical rationale behind the nonlinear teaching approach. Section 4 introduces the nonlinear teaching intervention in practice. Section 5 provides some reflections on the qualitative feedback and a formal quantitative evaluation of the nonlinear teaching practice using econometric models. Section 6 concludes.

## 2. Literature Review

Teaching large cohorts in higher education, particularly in quantitative fields like economics, presents distinct challenges and opportunities for pedagogical innovation. Traditional lecture methods often do not sufficiently engage students or accommodate diverse learning styles, which can impede the understanding and retention of complex economic concepts [17]. This review examines various teaching approaches designed for large classes and discusses the various indicators measuring the effectiveness and efficiency of teaching approaches.

### 2.1. Large-Cohort Teaching

Conventionally, large-cohort teaching has relied heavily on lecture-based methods, characterised by one-way communication from the instructor to the students [18]. This model has been criticised for promoting passive learning, where limited interaction can lead to decreased student engagement and poor retention [19]. In response, educational strategies have evolved to incorporate more interactive and student-centred approaches [20]. For example, the flipped classroom model reverses traditional expectations by requiring students to prepare before class, thus enabling active learning during lecture sessions through problem-solving and discussion [21]. Technology-enhanced learning tools, such

as clickers or online forums, have also been integrated to facilitate real-time feedback and increase student participation [22].

The effectiveness of large-cohort teaching is significantly enhanced when aligned with established learning theories [23]. Constructivist learning theory, which posits that learners construct their understanding and knowledge of the world through experiences and reflection on those experiences, supports environments that actively involve students in their learning [24]. Applying constructivist principles, instructors can facilitate learning by encouraging students to question, explore, and apply ideas in real-world contexts [25]. Experiential learning theory further supports this approach by emphasizing the role of experience in the learning process [26]. In large economics classes, these theories have been operationalised through case-based teaching where students analyse real economic scenarios, and through simulation games that mimic economic decision-making processes [27]. Research indicates that such practices not only improve conceptual understanding but also enhance students' ability to apply economic theories pragmatically [28].

Studies evaluating the impact of innovative teaching methods in large cohorts provide mixed results. Freeman et al. conducted a meta-analysis showing that active learning significantly increased student performance in science, engineering, and mathematics [19]. Similar studies in economics suggest that, while active learning techniques like the flipped classroom can improve examination scores, they require substantial adaptation by instructors and commitment from students [29]. Additionally, the use of technology in large classes has been shown to facilitate interaction and engagement but also presents challenges related to distraction and the digital divide [30].

After all, these innovative teaching approaches are only complementary to, not substitutes for, lectures [31]. Few attempts have been made to improve the effectiveness of learning and teaching within the traditional form of lectures. Essentially, the literature on large-cohort teaching mainly gets around the question by avoiding lectures, rather than resolving the problem per se. This paper, in contrast, aims to fill the gap in the literature directly. Hence, this paper aims to answer two research questions:

RQ1. How to make economics lectures match how we learn?
RQ2. How to evaluate the proposed nonlinear teaching approach?

### 2.2. Indicators for Evaluating Teaching Approaches

The effectiveness and efficiency of teaching approaches are paramount to achieving educational objectives. Therefore, indicators measuring effectiveness and efficiency are informative tools to evaluate different teaching approaches. There are three prevailing indicators in empirical literature.

First, student performance, often measured through marks, is a traditional and powerful indicator of the effectiveness of teaching methods. In economics education, where quantitative and analytical skills are emphasised, marks not only reflect the acquisition of subject-specific knowledge, but also critical thinking and problem-solving abilities. Studies have demonstrated a correlation between innovative teaching methods, such as case-based learning and simulation, and improved student performance in economics courses [32]. These findings suggest that methods that actively engage students tend to enhance their understanding and retention of complex economic theories [33].

Second, attendance rate is also frequently used as an indirect measure of student engagement and the attractiveness of the teaching approach. High attendance rates are often associated with more engaging lectures and a positive classroom environment, which are critical in subjects as challenging as economics. Research by Gupta and Pandey indicates that interactive lectures, where students participate in discussions and problem-solving activities, significantly boost attendance compared to traditional lecture-based sessions [34]. This correlation underscores the importance of interactive elements in maintaining student interest and engagement.

Third, student satisfaction surveys are a vital tool for assessing the quality of teaching and the overall student experience. These surveys typically evaluate various aspects of

the course and teaching methods, including the clarity of instruction, the relevance of the material, and the instructor's ability to inspire interest in the subject. In the field of economics, where theoretical concepts can seem detached from practical realities, teaching methods that effectively bridge this gap, such as real-world applications and experiential learning, tend to score higher on satisfaction metrics. A study by Ang et al. highlights that students rate courses higher when instructors successfully linked economic theories to current events and real-life economics issues [35].

Beyond marks, attendance, and student satisfaction, the literature on educational assessment recognizes several other indicators that can be used to measure the effectiveness and efficiency of teaching approaches. For example, graduation and retention rates are crucial indicators, particularly in higher education. Effective teaching methods should not only engage students, but also support them in successfully completing their courses and programs. Higher retention and graduation rates are often seen as indicators of successful teaching strategies that foster both academic and personal development among students [36]. In addition, as the ultimate aim of many educational programs, especially in higher education, is to prepare students for successful careers, the rate at which students gain employment in their field of study or advance in their careers can be an indicator of the effectiveness of teaching. This metric evaluates how well educational content and teaching methods prepare students for the professional world [37]. Moreover, feedback from alumni about the long-term impact of their education can provide valuable insights into the effectiveness of teaching methods. Alumni surveys might ask about the relevance of the skills they learned, their preparedness for professional challenges, and their overall educational satisfaction. This long-term perspective helps institutions to understand the enduring impact of their teaching methods [38]. Nevertheless, these indicators are more appropriate for the programme-level evaluation rather than lecture-level evaluation. As a result, we will adopt the three traditional indicators, i.e., marks, attendance, and satisfaction, in our empirical section. The next section will develop the proposed nonlinear teaching approach based on learning theories.

## 3. Conceptual Framework

The theoretical rationale of the nonlinear teaching approach is learning theories. The basic idea of this approach is to match the teaching and learning process and to promote the effectiveness and efficiency of disseminating and acquiring knowledge. There are several schools of thought on learning process. Learning theories provide a variety of conceptual frameworks to describe how information is absorbed, processed, and retained during learning. Cognitive, emotional, and environmental influences, as well as prior knowledge, all play a role in how knowledge and skills are acquired. There are two conventional learning theories shedding light on the proposed nonlinear teaching approach: Behaviourism and Constructionism.

### 3.1. Behaviourism

Behaviourism was coined by John Watson (1878–1959), who argued that learning is an aspect of conditioning, and advocated for a system of rewards and targets in education. In a simplified interpretation, Behaviourists model the learning process as stimulus—response–reinforcement [39]. If the lecturer can correctly provide some reward system to motivate the students, the effectiveness of the learning process may be promoted. Responses that result in favourable outcomes tend to be repeated and become established behaviour, while responses that lead to negative or neutral outcomes tend not to be repeated.

This learning-by-doing nature of the learning process implies that teaching process should provide appropriate stimulus from time to time. However, in a conventional "linear" teaching approach, as illustrated in Figure 1, the lecture contents usually start from simple things and become more and more difficult. The degree of complicatedness monotonically increases as the lecturer builds new knowledge on the previous elements. This linear fashion tends to make students frustrated (negative stimulus) in the middle of the module.

Once lost, always lost. It argues for a nonlinear design of the module contents in terms of difficulty, so that the students are rewarded by being able to understand (positive stimulus).

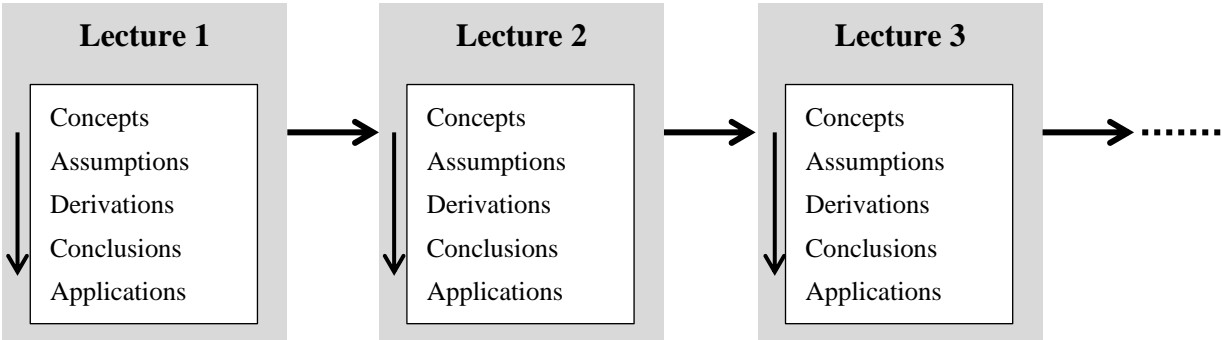

**Figure 1.** Linear Teaching Conceptual Framework.

There seems to be a conflict between the logical structure and the nonlinear design of the lecture contents. Logically speaking, the module should be designed accumulatively, i.e., later lectures should be built on the previous lectures. It inevitably leads to a simple-to-difficult linear fashion. However, this logical structure is an outcome of the learning process, rather than the learning process per se. No theory has been born as what it eventually looks like. There are detours and errors during the evolution of the theory, and the learning process is the same. Therefore, the teaching procedure does not have to be "logical" and precise in every stage. To provide a learning experience with positive stimulus, the lecturer can use some naïve interpretations (easy to understand but imprecise or even erroneous) in the first stage and correct them to a more precise definition later on (as illustrated in Figure 2). Moreover, nonlinear teaching is not from specific to general (as shown in linear teaching), but from general to specific in a holistic fashion. Of course, the lecturer must make the students aware that the initial interpretation is not precise and to be corrected and detailed later on. In this way, a module may have several lectures, in each of which we start with a simple and general understanding while continually correcting the initial understanding to a more precise level, so that the learners receive positive stimulus constantly while acquiring the knowledge. The idea behind this is that there is no shortcut in learning. Making mistakes is both natural and necessary for learning, and some appropriate detour may actually be more effective than linear progress.

To provide intuition, we revisit our example in the introduction again. To teach Sue to become familiar with the city, the local guide Joe may start like this. The city centre is very close to the Millennium stadium, which is very tall and easy to see from anywhere (initial stage). Follow the direction which leads you closer to that landmark and adjust when you are approaching the Millennium stadium (correcting). It is very likely that Sue will come across the city centre during the journey towards the stadium. Once this task (going to city centre) is accomplished, Joe can then teach Sue the next "lecture", say, going to national museum. A similar procedure can then be used.

However, Behaviourism does not account for free will and the internal prior knowledge of the learners. Therefore, the nonlinear teaching approach also draws some inspirations from Constructionism.

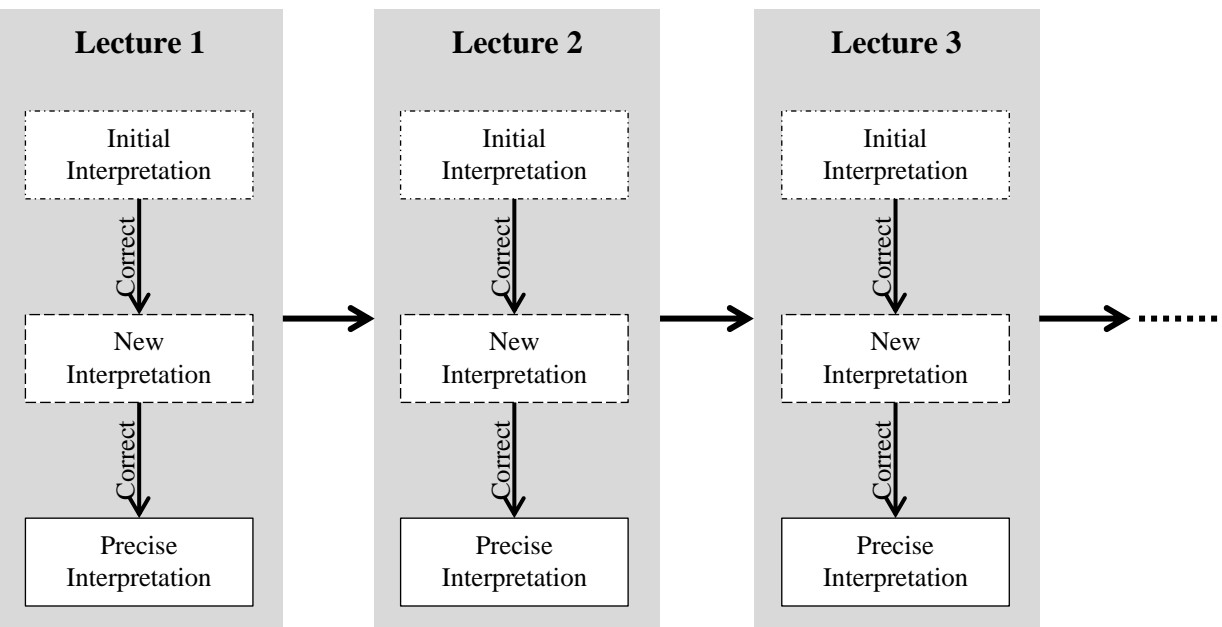

**Figure 2.** Nonlinear Teaching Conceptual Framework I.

*3.2. Constructionism*

Constructionism treats learning as a process of sense making, so the role of the instructor is as the facilitator. The two pioneer Constructionists, Piaget and Vygotsky, both appreciated the essence of internalising knowledge, rather than accepting the information as presented through rote-memory [40,41]. Everyone comes to the lecture with some existing knowledge, and the learning process is to accommodate the new information into the old system and framework. This implies that the lecturer needs to take into account the acceptability and conflicts between the contents of lecture with learners' prior knowledge.

This also implies that learning is not a one-way process. Rather, learners go back and forth between the new information and the existing knowledge. To match this nonlinear feature of learning process, lecturers need to bridge the gap between the learners' existing knowledge and the new knowledge. This connection should be built from time to time, not just once in the beginning of the lecture, because this feature of the learning process is continuous.

Let us still use the visitor's example, when Joe teaches Sue how to go to the national museum. Joe can always base his teaching on Sue's existing knowledge about the city. For example, now that Sue already knows how to go to the city centre, Joe can just tell Sue that the museum is just to the north of the city centre. The existing knowledge provides a good reference point to compare the new knowledge with.

One difficulty of associating with prior knowledge in teaching is that different students may have different backgrounds, and so, different prior knowledge. Sometimes, they do not even have prior knowledge, especially for introductory level modules. In this case, we can still use common sense or even metaphors to establish the bridge. It proves quite helpful for students to understand the abstract theories if lecturers have a good sense of humour. One explanation for this is because humour can link the complicated and dry theory to some vivid everyday experience which shares similar logic. For example, in macroeconomics, lecturers usually describe the relationship between the government and the public as a couple. The government may promise some policy during the election but follow a different policy afterwards (so-called "time inconsistency"). This can be likened to how a boy promises to love the girl during courting, but after establishing the relationship, the boy may change his behaviour. This sort of metaphor may greatly assist the students to understand the essence of theories, though the description is not precise. Another by-

product of using everyday examples to provide sense making is that students enjoy the learning process more and attendance could be improved.

Therefore, the sense making does not change the conceptual framework of the nonlinear teaching method developed above, but it provides an effective way of raising the initial interpretation of new knowledge. Usually, it can be a bird's eye-view of "where we are" from the whole picture. It is suggested to place the new knowledge into a bigger context, with which students are already familiar. Comparison and contrast with familiar contents can be used to help students accommodate the new knowledge in an existing framework. This can be supplemented by everyday examples or common sense to help students to grasp the essence of the theory. After this, a more systematic and precise exposition of the theory should be laid out and summarised.

### *3.3. Nonlinear Teaching Approach*

In the light of Behaviourism and Constructionism, I developed the general framework of the nonlinear teaching approach, as well as some techniques of implementing it. Other learning theories, such as social learning, may also contribute to enhancing the effectiveness and efficiency of teaching and learning. For example, assigning some roles for the students to solve some realistic problems. Group discussion can also be used to promote the exchange of knowledge and understanding in a multi-directional fashion, rather than linear one-directional flow from lecturer to students.

The nonlinear teaching approach can be formulated in a more precise version as in Figure 3, which is an upgraded and revised framework from Figure 2. The key difference is the review and reflection upon the system of knowledge along with the learning and teaching process, instead of just once at the beginning of each lecture and then a linear flow within each lecture. The purpose of revisiting "where are we" is to make sure the learners keep a close track of the teaching plan about what have been taught and what is to be taught. It can avoid the difficulty in finding their way once lost.

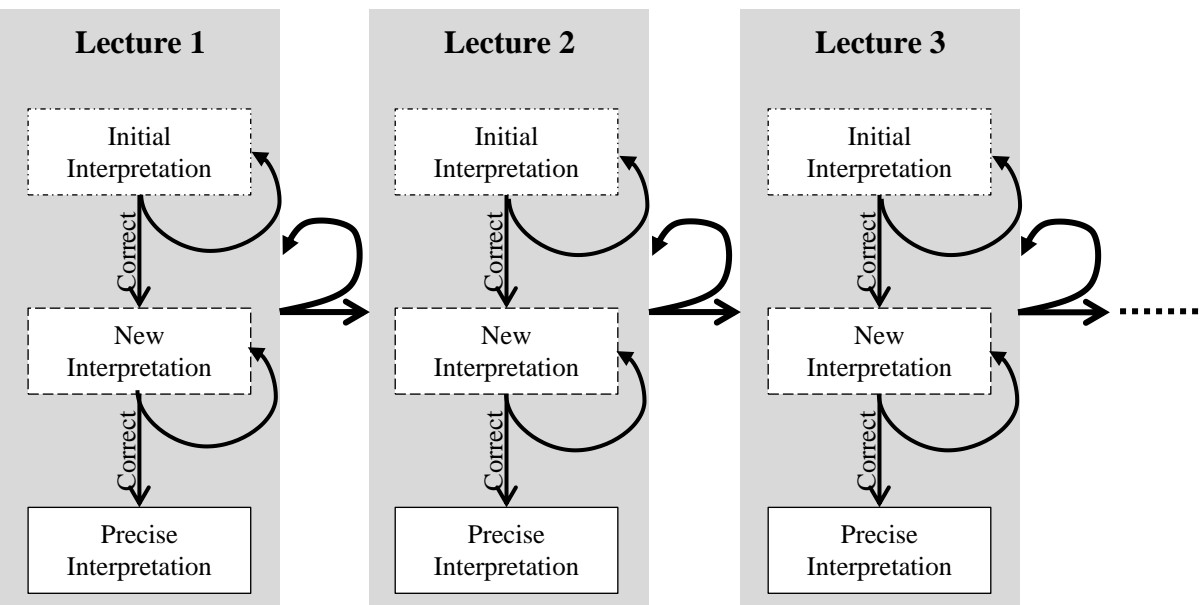

**Figure 3.** Nonlinear Teaching Conceptual Framework II.

One feasible and simple technique to realise this additional nonlinear feature in teaching is to show the outline after each section of each lecture. It could be even better if lecturers add a navigation panel in their slides. It is also advisable to revisit some important and/or difficult points, even if you think it is a bit repetitive. It is not repetitive to the learners!

Let us turn to our visitor's example once again. In addition to telling the route to Sue, Joe also gives Sue a GPS navigator, which can show her where she is in the city. That would greatly enhance the learning process. Even if Sue gets lost due to some confusing lanes, she will be able to get back to the right direction as long as she looks at the navigator. The outline in the lecture is like a static map, which helps, but the revision of the outline is like a dynamic GPS navigator, which helps a lot more.

To summarise, the nonlinear teaching approach is mainly derived from Behaviourism and Constructionism. The rationale is to match the nature of the learning process in teaching. We also discussed some techniques to implement nonlinear teaching including, but not restricted to:

- teaching from a rough to precise perspective;
- teaching from a general to specific perspective;
- sense-making, based on common knowledge and metaphors;
- role play and group discussion to promote active and interactive learning;
- a review of "where are we" at the beginning of each section and each lecture.

Note that nonlinear teaching does not mean simple repetition, but a spiral ascending process. The same contents may be recalled several times, but it should deepen the learners' understanding each time.

In fact, the readers may have already realised that this paper is written in a nonlinear teaching approach style. I started with an everyday life example (visitor) and have revisited this example several times. The illustration of the conceptual framework started with a simplified rough idea in Figure 2 and then became more precise by Figure 3. Imagine that if Figure 3 is shown directly upfront, some readers may feel confused due to the complicated lines and curves. A small step further with many steps may work much better than one big stride. If you have understood this paper well up to now, then the nonlinear teaching approach works on you.

## 4. Teaching Intervention in Practice

I implemented the nonlinear teaching approach in Intermediate Microeconomics, a large-cohort undergraduate core module every year in a British university. The following techniques were used.

First, in each lecture slide, I presented a navigation tree, showing where we are. I explicitly revisited the outline at the beginning of each section. Moreover, in the beginning of each lecture, I also revisited the topics covered and posited the current lecture in the big picture drawn in the first lecture. A simple revision was provided on each part when it was finished, but from a more general and higher perspective.

Second, everyday examples were elaborately designed to assist the students in accommodating the new theory into their existing knowledge system. For example, when I described "production function" in producer theory, a metaphor is used. I told the students to treat their exam marks as the "output" from the production function, and the students' effort as one input, while attendance as another input. More effort and a higher attendance tended to bring higher marks. A smart student with low inputs may still achieve a lower mark than another diligent student with higher inputs. The intelligence is the curvature of the production function, while the diligence is the inputs of the production function. Later on, this example can be used again to illustrate the concept of a "diminishing marginal product". For a given level of effort, a student's mark rises less and less if only one type of input rises.

Third, as mathematics is a main feature in Intermediate Microeconomics, such as differentiation and equation solving, to avoid confusing the students in the middle of the lecture, I always presented the intuition first, then the mathematical details, as well as a graphical illustration. More than one approach was presented to students, so that those with a weaker mathematical background could still follow the general argument without getting lost in the rest of the lecture.

Finally, group discussion was frequently used in seminars. Interactions among students were greatly encouraged. When one's role shifts from a learner to a discussant, students tend to be more active and understand the knowledge deeper in communication, because there is more than one angle of looking at the problems (Zhou, 2019). This method replaces uni-directional linear information flows from the lecturer to students (Figure 4A). Multi-directional nonlinear information flows among the lecturer, students, and other sources like the Internet and AI-based tools (Figure 4B).

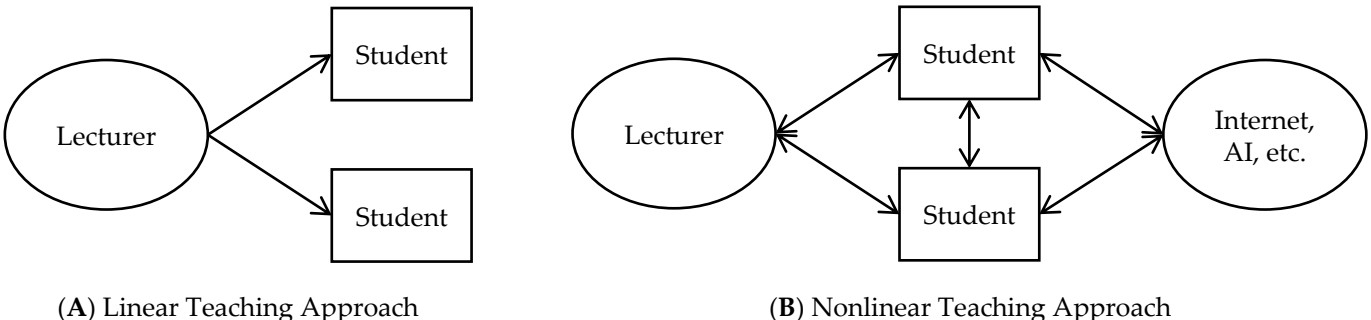

(**A**) Linear Teaching Approach          (**B**) Nonlinear Teaching Approach

**Figure 4.** Information flow of linear and nonlinear teaching approaches.

The next section will evaluate the effectiveness and efficiency of the nonlinear teaching approach using strict econometric models.

## 5. The Evaluation of the Approach

As a lecturer, I have access to the students' information, such as gender, race, age, programme, exam marks, and attendance for both current students (the "treated" group) and previous students (the "control" group). The evaluation of the nonlinear teaching intervention is based on two types of feedback. Conventional feedback was collected through questionnaires at the end of the semester before an exam. The response rate was 85.7%. However, qualitative evaluation based on a questionnaire is usually criticised as being biased because the students may fear that the lecturer can figure out who wrote what. Also, the answers to the questionnaire may only reflect what they think is true, not what is actually true. To make a consistent comparison between the "treated" students and the "control" students, I collected the second type of "feedback" by conducting a mock exam with the current students using the previous year's exam paper. This is like a natural experiment with the exogenous assignment of treatment, and it is guaranteed to compare like with like. One may argue that the current students may not be as well prepared when they take the mock exam as they are taking their actual exam. It could be handled by using a weighted average between their actual and mock exam marks. However, this remedy also has its drawbacks, for example what weight should be used. Since the mock exam is conducted about three weeks before the actual exam, it is not implausible to assume the students have a consistent performance. Therefore, it is complementary and superior to use the second type of feedback, based on their actual performance in their exam (or mock exam) marks. It is more objective and can provide a quantitative measure for effectiveness and efficiency. Therefore, I will briefly discuss the qualitative evaluation based on the first type of feedback and detail the quantitative evaluation based on the second type of feedback.

### 5.1. Qualitative Evaluation

The questionnaire asks the students two free-text questions:

(i)     What are you finding most useful about the module?
(ii)    What would improve the module?

For the first question, almost all students (94.4%) noted that the lectures were "well-structured" or "easy to follow", which implies that the navigation of the big picture

improved the students' learning experience. About a half of them (47.2%) mentioned that the lectures were closely relevant to "everyday life" and a "working environment", indicating that sense-making based on common knowledge worked well. Moreover, some students appreciated the multiple ways of presenting the same theory, including graphical, mathematical, and verbal. This nonlinear feature of the teaching approach also helps students from different backgrounds to understand the theory better and deeper.

Note that students were not aware that the lecturer carried out such a nonlinear teaching intervention before they filled in the questionnaire. This is to avoid biasing their opinions. Surprisingly, they seemed to identify the techniques of nonlinear teaching, and this is supportive evidence for the effectiveness of the teaching intervention.

For the second question, a main complaint is about the mathematics used in the module. Some business students do not have A-level maths, so they felt a bit challenged following the derivation of some mathematical models. This will be addressed by organising different seminar groups in the future.

*5.2. Quantitative Evaluation*

In addition to the qualitative evaluation, this paper also designs a formal econometric model to quantify the effectiveness and efficiency of the nonlinear teaching approach based on the second type of feedback. The mock exam marks of the current students are compared to the actual exam marks of the previous students, after controlling for other individual characteristics. There are two advantages of the teaching intervention. On the one hand, the outcome (the marks) is independent of the assignment of "treatment" by design, so it is a "natural experiment" (i.e., the data is experimental rather than observational). Thus, the treatment effect can be directly estimated without having to worry about endogeneity or a selection bias problem. On the other hand, the students have different attendance rates to lectures and seminars, which makes the "treatment" a continuous variable. Those who do not attend some of the lectures or seminars must review the contents themselves, so the teaching intervention has less effect on them. It is assumed that the students are occasionally absent because of random events, such as sickness or emergency, rather than a systematic lazy personality (In fact, the attendance is strictly monitored by the school and the students are required to provide justifiable reasons for absence. However, some students who consistently have low attendance (lower than 20%) are dropped from the analysis, because they may present a systematic difference in motivation). Therefore, the assignment of treatment is still exogenous, but this enables us to explore the treatment effect in more details. Nevertheless, the use of mock exams as a metric for learning outcomes assumes that students treat these mocks with the same seriousness as actual exams. If this is the case, then the students sitting in the mock exam should perform less well than if they are sitting in actual exams. The estimated treatment effect is a lower bound, which reinforces our argument, rather than weakens it.

If we treat the students' marks as the "output" of a "production function" of human capital, with students' attendance ($A_i$) as the "input". The "total factor productivity" of the production function can be affected by both the students' individual characteristics, including gender, race, native/overseas student, with/without A-level maths and business/economics students, as well as the lecturer's teaching approach (the teaching intervention dummy, or the "treatment", $T_i$). The implied econometric model can be written as:

$$marks_i = \alpha + \mathbf{ind}_i \cdot \boldsymbol{\beta} + \gamma \times A_i + \delta \times T_i + \phi \times A_i \times T_i + \varepsilon_i$$

The intercept $\alpha$ is the average level of productivity of an average student, and the vector $\boldsymbol{\beta}$ captures the effects of individual characteristics ($\mathbf{ind}_i$) on productivity. The coefficient $\gamma$ can be interpreted as the return to attendance, i.e., how many marks an average student could improve by one more attendance. The coefficient $\delta$ is the "treatment effect" of the teaching intervention, because only the "treated" students ($T_i = 1$) have this shifting term in their productivity (intercept effect). Finally, there is also a cross-product term (the attendance rate $A_i$ multiplied by the intervention dummy), and the coefficient $\phi$ is the effect

of the treatment on the return to attendance (slope effect). If a student is exposed to the nonlinear teaching approach, then their return to attendance is equal to $\gamma + \phi$, rather than $\gamma$. The coefficient $\delta$ can be used to evaluate the effectiveness, while $\phi$ can be used to evaluate the efficiency, because the different degrees of exposure to the teaching intervention also improve learning.

In total, there are 342 students registered in the treated group, including both economics students (78%) and business students (12%). Three students have consistently low attendance and are dropped in the analysis. The control group has 316 students with a similar structure, but all students are included. The distribution of marks and attendance of the two groups are contrasted in Figure 5. It is shown that the distribution of marks of the treated group is more concentrated and more towards higher marks, and so is the distribution of attendance.

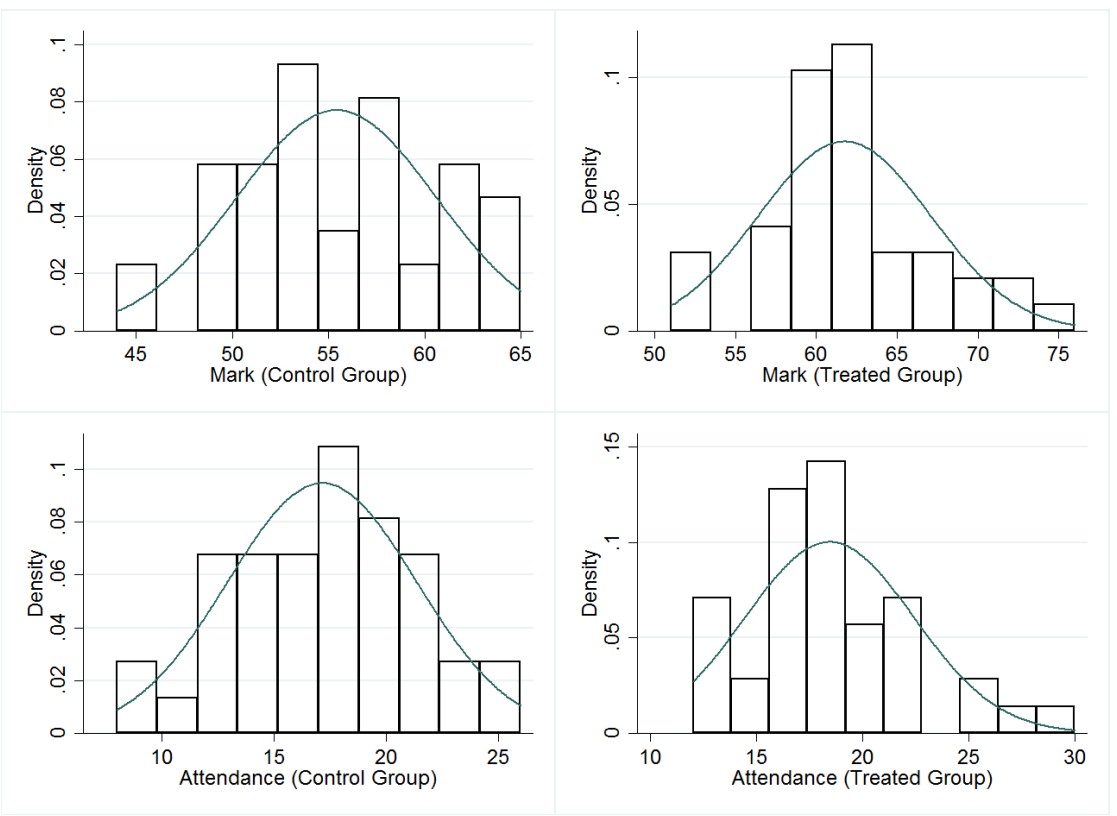

**Figure 5.** Distribution of Marks and Attendance.

The regression results shown in Table 1 are a stricter way to verify the observed changes in distributions of marks and attendance. The first two columns of the table focus on evaluating the effectiveness and efficiency of the nonlinear teaching approach using two alternative model specifications (both are estimated using OLS with robust standard errors against possible heteroskedasticity), and the last column attempts to find out if the new teaching intervention improves the attendance (Poisson regression is used because the dependent variable is a count variable). To summarise, the three hypotheses to be tested are:

**Hypothesis (H1).** *The nonlinear teaching approach increases the effectiveness of learning and teaching, or a positive average mark, i.e., $\delta > 0$.*

**Hypothesis (H2).** *The nonlinear teaching approach increases the efficiency of learning and teaching, or a higher return to attendance, i.e., $\phi > 0$.*

**Hypothesis (H3).** *The attendance rate is improved because of H1 and H2.*

Both model specifications imply a significant and positive treatment effect of the nonlinear teaching approach, so H1 is verified. If we do not include the cross-product term $A_i \times T_i$ as in the first model specification, then the treatment effect (the coefficient of $T_i$) is $\hat{\delta} = 7.428$. In other words, the students' performance (marks) is improved by more than seven marks on average. However, if the cross-product term is included, as in the second specification, the treatment effect reduces to $\hat{\delta} = 4.171$, but the return to attendance increases from 0.469 to $\hat{\gamma} + \hat{\phi} = 0.384 + 0.182 = 0.566$. This finding verifies H2. Based on the two regressions, we can actually decompose the overall treatment effect into two components: (i) the improvement purely due to a more effective teaching practice, accounting for $\frac{4.171}{7.428} = 56.15\%$ of the treatment effect; and (ii) the improvement due to the indirect effect on return to attendance, i.e., efficiency improvement, accounting for the rest, 43.85%.

**Table 1.** Estimation Results.

| Dep. Var. | Mark | Mark | Attendance (A) |
|---|---|---|---|
| male | −0.511 | −0.59 | −0.04 |
| | (0.462) | (0.456) | (0.06) |
| white | 0.471 | 0.464 | 0.037 |
| | (0.548) | (0.512) | (0.067) |
| native | 1.062 ** | 1.021 ** | 0.005 |
| | (0.446) | (0.468) | (0.066) |
| A-level | 9.147 *** | 9.138 *** | 0.022 |
| | (0.431) | (0.416) | (0.061) |
| business | −4.739 *** | −4.775 *** | −0.011 |
| | (0.528) | (0.518) | (0.07) |
| $A_i$ | 0.469 *** | 0.384 *** | |
| | (0.048) | (0.053) | |
| $T_i$ | 7.428 *** | 4.171 *** | 0.078 |
| | (0.407) | (1.545) | (0.057) |
| $A_i \times T_i$ | | 0.182 ** | |
| | | (0.088) | |
| _cons | 44.132 *** | 45.684 *** | 2.828 *** |
| | (0.915) | (1.028) | (0.099) |
| Method | Robust OLS | Robust OLS | Poisson |
| R-sq | 0.926 | 0.93 | |
| adj. R-sq | 0.919 | 0.922 | |
| pseudo R-sq | | | 0.007 |
| AIC | 323.482 | 321.519 | 462.419 |
| BIC | 342.538 | 342.957 | 479.093 |

Notes: Standard errors in the parentheses, ** 5%, *** 1%.

In particular, the efficiency improvement can be quantified by the coefficient of the cross-product term relative to that without treatment: $\frac{0.182}{0.384} = 47.40\%$. This significantly higher return to attendance should, in principle, encourage a higher attendance, because their marks can be improved more efficiently by attending the lectures (H3). This hypothesis is tested by a Poisson regression (the last column), but it turns out that the teaching intervention does not significantly increase the students' attendance. This is perhaps because the students are not aware of the implementation of the teaching intervention while it was implemented.

Other interesting findings from the regressions include:

- Gender and race do not significantly contribute to different marks.
- Native students tend to perform better than international students, maybe because of a language advantage.
- Students with A-level maths are expected to perform better than those without.

- Students enrolled in business programmes tend to obtain lower marks than those enrolled in economics programmes.

## 6. Conclusions

This paper proposes a nonlinear teaching approach in the context of teaching economics, though it is ready to be generalised to other disciplines. The rationale is derived from two learning theories, Behaviourism and Constructionism. By fitting the nature of the learning process, the nonlinear teaching approach is expected to improve the effectiveness and efficiency of teaching. A main feature of nonlinear teaching approach is to review and associate the new knowledge with the existing framework, either via sense making or via progressive repetition. It lies between the conventional passive teaching and the unstructured interactive teaching. Therefore, nonlinear teaching is more appropriate to natural science and quantitative social science like economics.

Some specific techniques are discussed and applied to a specific scenario in an undergraduate teaching practice. The teaching intervention received positive feedback in both qualitative and quantitative evaluation. Formal regression analysis verifies the effectiveness and efficiency of the nonlinear teaching approach based on the natural experiment (H1 and H2). It is estimated that 56.15% of the overall improvement on student performance can be attributed to the more effective teaching approach (an intercept effect), and the other half is derived from a higher return to attendance (a slope effect). If the students were aware of this, they would have improved their attendance, but this hypothesis (H3) is not supported in the Poisson regression.

One possible shortcoming of nonlinear teaching approach, as reflected by the student feedback, is that too much repetition may make the students bored and less receptive. This might explain why attendance was not improved significantly, because the negative effect cancels out the positive effect on the attractiveness of the lectures. Using the visitor's example again, if Sue goes through the same roads in Cardiff hundreds of times, she may lose her joy of travelling—which turns to boring commuting. If there is less passion, then there is less joy of travelling. Therefore, lecturers who adopted nonlinear teaching may have to think of a way of promoting the students' motivation and interest while benefitting from the improvement in effectiveness and efficiency.

It is also worth noting that there is no one-size-fits-all solution in teaching. Different types of modules require different types of teaching approaches. For example, for introductory modules like introduction to economics, teaching approaches often emphasize clarity, simplicity, and motivation. For advanced modules like microeconomic theory and macroeconomic theory, teaching approaches may need to be more structured. The proposed nonlinear teaching approach is more appropriate for the latter. The approach has been supported in a core module of undergraduate economics programmes, but its principles can be applicable to other subjects with similar features—abstract concepts and quantitative contents. In general, with the fast development of AI technology, the role of lecturers must evolve from a knowledge dispenser to a paradigm navigator.

**Funding:** The APC was funded by Open Access Support Team, Cardiff University.

**Institutional Review Board Statement:** Ethical review and approval were waived for this study because student feedback used in this paper was part of regular teaching and learning activities.

**Informed Consent Statement:** Informed consent was obtained from all subjects involved in the study.

**Data Availability Statement:** Data and codes are available on request.

**Conflicts of Interest:** The author declares no conflict of interest.

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
