# Peer review of "Make Lectures Match How We Learn: The Nonlinear Teaching Approach to Economics"

_education, doi:10.3390/educsci14050509_

Round 1

Reviewer 1 Report

Comments and Suggestions for Authors

This is a great article that will help in my teaching.  

There is two issues with the manuscript, and it may have to do with how it was formatted for review.  

1) Wherever the author references the figures an error appears instead: "Error! Reference source not found."  This appears in lines 157, 176, 235, 251, 252, 286, 287, 

2) The boxes in Figures 1 and 2 seem to be cropped.  And what are the dots that the last arrow on right mean?

In terms of comments, I do have a few.

1) Given this method of teaching, would it be possible to cover all topics in a semester?  For a principals course, I must cover a considerable amount of topics which may not allow for the style proposed by the author.  I understand the solution may be to reduce concepts, but many are still necessary for students that wish to pursue a degree in Economics.

2) Related to 1), I teach diverse group of students across many fields, including engineering, physics, business, political science, as well as students that have not declared a major.  From the results, business students have the lowest outcomes.  It would be interesting to know how this method would work with a more diverse group.

3) Do the authors have information on grades from a previous semester to include as an individual characteristic?  My issue with math is that it depends, in part, on the type of major the student is pursuing and the grades they have earned to date.  I would be interested to see how their academic standing affects their performance in the course given the intervention.

Author Response

I am very grateful for the thoughtful and constructive comments from the reviewer. Here are my improvements following your suggestions. I will present my point-to-point responses below.

This is a great article that will help in my teaching. There is two issues with the manuscript, and it may have to do with how it was formatted for review.  

1) Wherever the author references the figures an error appears instead: "Error! Reference source not found."  This appears in lines 157, 176, 235, 251, 252, 286, 287.

Thanks for pointing these out. I have now corrected these errors in cross-references of the orignal text.

2) The boxes in Figures 1 and 2 seem to be cropped.  And what are the dots that the last arrow on right mean?

The format of the figures has been changed. I have now re-formatted them to make sure they display correctly. The dots on the right end means other lectures to continue. I have now changed the shapes to avoid misunderstanding.

In terms of comments, I do have a few.

1) Given this method of teaching, would it be possible to cover all topics in a semester?  For a principals course, I must cover a considerable amount of topics which may not allow for the style proposed by the author.  I understand the solution may be to reduce concepts, but many are still necessary for students that wish to pursue a degree in Economics.

Thanks. You have pointed out the boundary condition for the teaching approach. We have added the following in our discussion:

Indeed, there is no one-size-fits-all solution. Different types of modules require different types of teaching approaches. For example, for introductory modules like introduction to economics, teaching approaches often emphasize clarity, simplicity, and motivation. For advanced modules like microeconomic theory and macroeconomic theory, teaching approaches may need to be more structured. The proposed nonlinear teaching approach is appropriate for this type of modules. 

2) Related to 1), I teach diverse group of students across many fields, including engineering, physics, business, political science, as well as students that have not declared a major.  From the results, business students have the lowest outcomes.  It would be interesting to know how this method would work with a more diverse group.

As a quantitative subject, economics can be challenging to those without mathematical foundation. Economics students tend to have better mathematical background than business students. One indicator is the proportion of them having A-level maths. The proposed nonlinear teaching approach makes full use of common sense to compensate for the unbalanced mathematical background, so it is a good scenario to apply the approach.

3) Do the authors have information on grades from a previous semester to include as an individual characteristic?  My issue with math is that it depends, in part, on the type of major the student is pursuing and the grades they have earned to date.  I would be interested to see how their academic standing affects their performance in the course given the intervention.

You are right that individual characteristics can be informative. Unfortunately, according to the data protection policy in the university,  individual data are not available for all modules. We only have access to  previous marks of a particular module. To partly correct for that, we do have controlled the fixed effects by the programmes students are enrolled.

Reviewer 2 Report

Comments and Suggestions for Authors

The paper presents an innovative teaching strategy named the "Nonlinear Teaching Approach," aimed at enhancing learning efficiency in undergraduate economics education. It assesses the approach's effectiveness through a mixed-methods evaluation comprising both qualitative and quantitative analyses.

This review suggests areas for improvement:

-         The study is confined to a single module in an undergraduate economics course, limiting its generalizability to other disciplines or educational contexts. Please discuss this.

-         The paper provides an overview of the econometric model but lacks detailed justification for the choice of variables and the model's assumptions.

-         The approach did not significantly improve student attendance, a key component of engagement and learning outcomes. Investigating and addressing the factors influencing attendance directly could enhance the approach's effectiveness. Please discuss this.

-         The evaluation primarily focuses on exam performance as the metric for learning outcomes, which may not fully capture the breadth of learning and skill development.

-         There is limited discussion on the challenges faced in implementing the nonlinear teaching approach and strategies for overcoming these challenges. Providing detailed insights into the practical challenges encountered, along with recommendations for instructors looking to adopt this teaching method, would greatly benefit readers and potential adopters.

-         Creating detailed guidelines and resources for instructors could facilitate the adoption of the nonlinear teaching approach.

-         The study assumes comparability between the treated and control groups based on previous year’s performances. However, inherent differences between cohorts, could influence outcomes.

-         Explore alternative methods of assessing learning outcomes that do not rely on exam performance alone, such as project-based assessments.

-         Some figures are not clear.

-         Some errors in referencing figures inside the text.

-         Discuss more in detail the limitations of the study.

-         The use of mock exams as a metric for learning outcomes assumes that students approach these with the same seriousness as actual exams. This assumption may not hold, affecting the reliability of this measure.

Comments on the Quality of English Language

No comments

Author Response

I am very grateful for the thoughtful and constructive comments from the reviewer. Here are my improvements following your suggestions. I will present my point-to-point responses below.

The study is confined to a single module in an undergraduate economics course, limiting its generalizability to other disciplines or educational contexts. Please discuss this.

Thanks for pointing out the limitations. We have now added a discussion paragraph on this point in the conclusion section. Indeed, there is no one-size-fits-all solution. Different types of modules require different types of teaching approaches. For example, for introductory modules like introduction to economics, teaching approaches often emphasize clarity, simplicity, and motivation. For advanced modules like microeconomic theory and macroeconomic theory, teaching approaches may need to be more structured. The proposed nonlinear teaching approach is more appropriate for the latter.

The paper provides an overview of the econometric model but lacks detailed justification for the choice of variables and the model's assumptions.

We have now justified the econometric model by the economic logic as follows. If we treat the students’ marks as the “output” of a “production function” of human capital, with students’ attendance (A) as “input”. The “total factor productivity” of the production function can be affected by both the students’ individual characteristics, including gender, race, native/overseas student, with/without A-level maths and business/economics students, as well as the lecturer’s teaching approach (the teaching intervention dummy, or the “treatment”, T). 

The approach did not significantly improve student attendance, a key component of engagement and learning outcomes. Investigating and addressing the factors influencing attendance directly could enhance the approach's effectiveness. Please discuss this.

We discussed the finding on attendance in the conclusion section. One possible shortcoming of nonlinear teaching approach, as reflected by the student feedback, is that too much repetition may make the students bored and less sensitive. This might explain why attendance was not improved significantly, because the negative effect cancels out the positive effect on the attractiveness of the lectures. Using the visitor’s example again, if Sue goes through the same roads in Cardiff hundreds of time, she may lose her joy of travelling—which turns to boring commuting. If there is less passion, then there is less joy of travelling. Therefore, lecturers who adopted nonlinear teaching may have to think of a way of promoting the students’ motivation and interest, while benefitting from the improvement in effectiveness and efficiency.

The evaluation primarily focuses on exam performance as the metric for learning outcomes, which may not fully capture the breadth of learning and skill development.

This is a very good point. To provide complementary evidence other than exam performance, we did a qualitative evaluation by questionnaire. The free texts in the feedback show further support to the nonlinear teaching approach. Details can be found in subsection 4.1.

There is limited discussion on the challenges faced in implementing the nonlinear teaching approach and strategies for overcoming these challenges. Providing detailed insights into the practical challenges encountered, along with recommendations for instructors looking to adopt this teaching method, would greatly benefit readers and potential adopters.

We have now elaborated the challenges (too much repetition for some students, so the efficiency is low) and limitations (suitable for advanced modules, not for introductory modules) in the conclusion section.

Creating detailed guidelines and resources for instructors could facilitate the adoption of the nonlinear teaching approach.

We have added four guidelines in section 3 for instructors to facilitate the adoption: (1) reappearing navigation tree, (2) common sense making, (3) from rough to precision, (4) multi-directional communication.

The study assumes comparability between the treated and control groups based on previous year’s performances. However, inherent differences between cohorts, could influence outcomes.

This comment is true for all observational studies. Fortunately, for university recruitment, our students are selected based on their academic background every year without significant differences in the student quality. Therefore, the students in two adjacent years are generally comparable. Furthermore, we use control variables such as gender, race, nationality, A-level maths, and programmes, so that the estimated treatment effect is identified. Other unobserved characteristics such as IQ and personality are arguably randomly distributed, which should not systematically bias our estimates.

Explore alternative methods of assessing learning outcomes that do not rely on exam performance alone, such as project-based assessments.

This is a good thought. However, introducing new assessments only works for the current cohort (the treated group). There is no way of travel back in time to get the control group to do the same assessments for comparison. As a complement, we use free text questionnaire to collect qualitative feedbacks.

Some figures are not clear.

All figures are now revised to a high quality.

Some errors in referencing figures inside the text.

All cross-references are now corrected.

Discuss more in detail the limitations of the study.

Done in the conclusion section.

The use of mock exams as a metric for learning outcomes assumes that students approach these with the same seriousness as actual exams. This assumption may not hold, affecting the reliability of this measure.

This is a fair point that the use of mock exams as a metric for learning outcomes assumes that students approach these with the same seriousness as actual exams. If this is the case, then the students sitting in the mock exam should perform less well than if they are sitting in actual exams. The estimated treatment effect is a lower bound, which reinforces our argument rather than weakens it. We have added this paragraph in section 4.2.

Reviewer 3 Report

Comments and Suggestions for Authors

Please describe the research gap clearly, describe more detail figure 1, figure 2 and figure 3. Please explain the population sample and time period, and more detail the indicators measuring the  effectiveness and efficiency then describe the specific finding to show the novelty.

Author Response

Please describe the research gap clearly.

We have elaborated the research gap in the pedagogy literature in the introduction section. First, we point out that "the teaching approach in higher education institutions seems to evolve slowly, especially for those in quantitative subjects like economics (Becker and Watts, 1996, 2001)." Then, we focus on the current gap in pedagogy of economics. "To adapt to these disruptive challenges, blended teaching has become popular in higher education (Zheng & Lee, 2023). Nevertheless, it is usually recognised that existing pedagogies do not always fit students’ learning process (Rodriguez et al., 2021)." Building on the critical review, we raised the essence of the gap, i.e., lectures do not match how we learn: "an important feature of learning process is nonlinearity—learners do not acquire new knowledge in a one-way fashion."

Describe more detail figure 1, figure 2 and figure 3.

The three figures have been replotted for better exposition and explanation.

Please explain the population sample and time period, and more detail the indicators measuring the effectiveness and efficiency then describe the specific finding to show the novelty.

The sample size N = 324 for the treated group and N = 316 for the control group. Effectiveness is measured by the regression coefficient of the treatment T (taught by the nonlinear teaching approach). Efficiency is measured by the interactive term between the exposure to the treatment (attendance rate) and the treatment per se. This design is a well established econometric technique known as shift-share treatment--the "shift" is the treatment, and the "share" is the exposure. Based on the two regressions, we can actually decompose the overall treatment effect into two components: (i) the improvement purely due to a more effective teaching practice, accounting for 56.15% of the treatment effect; and (ii) the improvement due to the indirect effect on return to attendance, i.e. efficiency improvement, accounting for the rest 43.85%. 

Round 2

Reviewer 2 Report

Comments and Suggestions for Authors

I thank the authors for their efforts. However, I believe that they did not address my comments as thoroughly as they should have. My comments were intended to significantly improve the quality and usefulness of the paper for its readers. Despite my previous comments, only three small paragraphs were added. Additionally, there is no specific literature review section. Either create a distinct section that clearly outlines, the existing literature, the research gaps and questions, or clearly include this information in the introduction and rename it "Introduction and Literature Review". It is crucial that the literature review, gaps, and research questions are clearly presented. I really count on the authors to thoroughly improve the paper, rather than making the superficial changes seen in the previous revision.

Comments on the Quality of English Language

No comments.

Author Response

I am much grateful for the important comments and another opportunity to improve the paper. I take the comments seriously and have written a separate section on literature review (new section 2).

This section covers two themes, one on large cohort teaching (subsection 2.1) and the other on indicators measuring effectiveness and efficiency of teaching approaches. Both classical references and recent ones are used to base the review.

I also follow the suggestions to clearly state the gap in the literature and the two research questions (RQ1 and RQ2).

Furthermore, the review is closely linked to the theoretical framework and the empirical evaluation that follow.

Reviewer 3 Report

Comments and Suggestions for Authors

Please refer to literature for the indicators measuring the effectiveness and efficiency. 

Author Response

(The authors gave the same response as above.)

Round 3

Reviewer 2 Report

Comments and Suggestions for Authors

Problems with section numbering after you added the new section.

Comments on the Quality of English Language

No comments